# Effectiveness of a Water Disinfection Method Based on Osmosis and Chlorine Dioxide for the Prevention of Microbial Contamination in Dental Practices

**DOI:** 10.3390/ijerph191710562

**Published:** 2022-08-24

**Authors:** Michele Totaro, Federica Badalucco, Francesca Papini, Niccolò Grassi, Marina Mannocci, Matteo Baggiani, Benedetta Tuvo, Beatrice Casini, Giovanni Battista Menchini Fabris, Angelo Baggiani

**Affiliations:** 1Department of Translational Research and New Technologies in Medicine and Surgery, University of Pisa, 56126 Pisa, Italy; 2San Rossore Dental Unit, 56122 Pisa, Italy

**Keywords:** dental unit waterlines, reverse osmosis, chlorine dioxide, waterborne pathogens

## Abstract

In dental clinics, the infections may be acquired through contaminated devices, air, and water. Aerosolized water may contain bacteria, grown into the biofilm of dental unit waterlines (DUWLs). We evaluated a disinfection method based on water osmosis and chlorination with chlorine dioxide (O-CD), applied to DUWL of five dental clinics. Municipal water was chlorinated with O-CD device before feeding all DUWLs. Samplings were performed on water/air samples in order to research total microbial counts at 22–37 °C, *Pseudomonas aeruginosa*, *Legionella* spp., and chlorine values. Water was collected from the taps, spittoons, and air/water syringes. Air was sampled before, during, and after 15 min of aerosolizing procedure. *Legionella* and *P. aeruginosa* resulted as absent in all water samples, which presented total microbial counts almost always at 0 CFU/mL. Mean values of total chlorine ranged from 0.18–0.23 mg/L. Air samples resulted as free from *Legionella* spp. and *Pseudomonas aeruginosa.* Total microbial counts decreased from the pre-aerosolizing (mean 2.1 × 10^2^ CFU/m^3^) to the post-aerosolizing samples (mean 1.5 × 10 CFU/m^3^), while chlorine values increased from 0 to 0.06 mg/L. O-CD resulted as effective against the biofilm formation in DUWLs. The presence of residual activity of chlorine dioxide also allowed the bacteria reduction from air, at least at one meter from the aerosolizing source.

## 1. Introduction

Water output from dental unit waterlines (DUWLs) is often contaminated with high densities of environmental microorganisms [1,2,3].

High microbial levels have been found in output water from handpieces and air/water syringes [4]. The suck-back of biological fluids from the oral cavities of patients (back-contamination) was also reported as an important cause of DUWLs contamination [5].

The presence of small narrow-bore hydrophobic polymeric plastic tubing, electrical components that can heat the water (20–25 °C), and the discontinuous and low water flow are all factors that contribute to microbial growth and biofilm formation [1].

In particular, the formation of biofilms presents difficulties in maintaining clean DUWLs during routine dental practice [2].

Many contaminating microbial genera have been isolated and identified in water samples collected from dental units; *Pseudomonas* spp. and *Legionella* spp. are the prevalent bacteria found [6].

DUWLs may also be important replication sites for free-living amoebae and protozoa that enable the maintenance of pathogenic intracellular bacteria, increasing their resistance to disinfectants [5].

Considering this contamination, dental units are recognized as a potential source of infection for human health (Legionnaires’ disease, wound infections caused by *Pseudomonas* spp., etc.), especially threatening dentists and patients, who are regularly exposed to water and water–aerosol emitted by aerosol-generating procedures [7,8]. Moreover, the risk of infections is increased by high microbial load of aerosols derived from saliva, blood, nasopharyngeal secretions, plaque, calculus, and dental materials [9]. Bacteria including *Mycobacterium* spp., *Pseudomonas* spp., *Legionella* spp., *Staphylococcus aureus*, and *Streptococcus* spp., and viruses, such as rhinovirus, HIV, HBV, HCV, and herpes viruses, may be carried through airborne particles [9].

In particular, the current diffusion of the SARS-CoV-2 pandemic has highlighted the risk of potential infectivity of dental aerosols [10].

Previous studies have focused on the effectiveness of numerous disinfectants used for the cleaning and maintenance of DUWLs [2,11,12].

Reverse osmosis associated with chlorine dioxide treatment belongs to a series of automated patented equipment for the treatment and disinfection of water, applicable in dental units, and capable of supplying dental units, autoclaves, and the sterilization room sink through an additional tap installed on the same.

The principle of disinfection is characterized by two basic procedures: reverse osmosis applied to the inlet water of the unit (which will be demineralized and free of microbial load) and continuous disinfection of water with chlorine dioxide (which is the real active principle of the system) [13].

Some studies [14] report that SARS-CoV is susceptible to disinfectants in water such as chlorine dioxide and is completely deactivated at a concentration and in times lower than those required to break down the concentrations of traditional bacterial indicators of fecal contamination, such as the abetment of 1 × 10^7^ CFU/mL *Escherichia coli* in 5 min with 0.2 mg/L chlorine dioxide [15], which may be evaluated by the standard method ISO 20776-1:2019 [16].

The aim of this study was to evaluate the effectiveness of the disinfection method based on water osmosis and chlorination, with chlorine dioxide (O-CD) applied to the DUWL of a dental practice.

## 2. Materials and Methods

The study was performed from January to May 2021 in a dental practice with five clinics.

Informed consent of the observational study was submitted to all healthcare workers of the dental practice. All samplings were performed in the absence of patients and the aerosolizing dental procedures have only been simulated.

Each clinic (from 15 to 25 m^2^) has one dental unit and a tap. Only one 25 m^2^ clinic (C5) is dedicated to oral surgical activities, while the other four clinics (C1, C2, C3, C4) are used for ambulatory activities.

The whole water network is fed by municipal water, which is chlorinated with O-CD device (Figure 1) after desalination with a reverse osmosis device. Each dental unit and taps receive desalinated and chlorinated water.

Overall, three microbiological sampling plans were carried out on air and water matrices (January, March, and June 2021), with a total of 42 samples.

For each dental unit, 2 L of water were collected from the spittoon and handpieces (air/water syringes). 2 L of water were also collected from taps. Residual and total chlorine concentration were measured in all samples.

Water analyses were performed in order to enumerate the total count microbial at 22 and 37 °C (ISO 6222:1999) [17], *Pseudomonas aeruginosa* (ISO 16266:2008) [18], *Legionella* spp. (ISO 11731:2017) [19], as described by Council Directive 98/83/EC [20] and Italian Guidelines for Legionnaires’ disease prevention and control [21].

Air sampling was performed in three different moments:At rest, before clinical activities.During aerosolization, which was performed by handpieces and air/water syringes use for 15 min.At rest, before aerosolization procedures.

Air sampling (45 samples) was performed as described by Italian INAIL guidelines [22], using the Microflow 90 (Aquaria, Italy), which was located on dental chair (patient chair), one meter from the aerosolization source. From each air sampling, 500 L of air (flow rate of 120 L/min) were aspired for the research of the total microbial counts at 22 and 37 °C, *Pseudomonas aeruginosa*, and *Legionella* spp. Total chlorine concentration were measured in all air sampling point using Quantofix Chlorine Indicators.

After air samplings, all plates were incubated as described below:Plate Count Agar for detection of total microbial counts at 22 and 37 °C in 72 and 48 h, respectively;Cetrimide Agar for *Pseudomonas aeruginosa* detection in 48 h;Buffered Charcoal Yeast Extract (BCYE) Agar for *Legionella* spp. detection in 10 days in jars under an atmosphere containing 2.5% CO_2_.

Statistical analyses were performed in order to verify normality of distributions by the Kolmogorov–Smirnov test. For values detected from air samples, the Kruskal–Wallis test and Dunn’s test were used to compare the total microbial counts at 22 and 37 °C detected in pre-aerosolization, aerosolization, and post-aerosolization conditions. The power tests were carried out to estimate the sample sizes. The 1-beta values of the significant variables were >0.8, proving acceptable sample sizes.

Correlation tests were performed, and Pearson’s coefficients were calculated with the aim of analyzing the correlations between airborne bacteria reduction and chlorine values from air during aerosolizing procedure. 95% confidence levels were defined for the statistical tests. Therefore, we considered the following ranges of values: |0–0.3| (weak correlation); |0.3–0.7| (moderate correlation); |0.7–1| (strong correlation). The statistical analysis was fulfilled using the IBM SPSS software package.

## 3. Results

Microbiological analysis, performed on water samples collected from all five clinics, highlights the absence of *Pseudomonas aeruginosa* (0 CFU/100 mL) and *Legionella* spp. (<100 CFU/L). Mean values of microbial counts at 22 °C ranged from 0 to 3 CFU/mL, while the total microbial counts at 37 °C were always 0 CFU/mL. Moreover, all mean free and total chlorine values were compliant with the recommended value (0.2 mg/L by the European Regulation 98/83/CE). Mean values of free chlorine ranged from 0.17 to 0.22 mg/L, while mean values of total chlorine ranged from 0.18 to 0.23 mg/L.

The highest chlorine values were obtained from water collected by taps (from 0.18 to 0.23 mg/L), while a weak chlorine reduction was observed in samples collected from spittoons and handpieces (from 0.17 to 0.22 mg/L).

Table 1 shown the mean values of microbiological analysis performed for water samples. Considering the absence of handpieces in dental clinic 5, no data are recorded.

Microbiological results obtained from air showed total microbial counts at 22 and 37 °C higher than 1 × 10^2^ CFU/m^3^ in pre-aerosolization conditions (mean values ranged from 1.2 × 10^2^ ± 5 × 10 CFU/m^3^ to 3.4 × 10^2^ ± 9 × 10 CFU/m^3^). Overall, a mean total microbial count of 2.1 × 10^2^ ± 1 × 10^2^ CFU/m^3^ was detected in air samples collected after the aerosolizing procedures. In this condition, no chlorine traces were revealed in air.

During the 15 min of aerosolization, total microbial counts at 22 and 37 °C ranged from 7.6 × 10 ± 2.4 × 10 CFU/m^3^ to 2 × 10^2^ ± 4.3 × 10 CFU/m^3^, with a mean value of 1.4 × 10^2^ ± 4 × 10 CFU/m^3^. A mean total chlorine value of 0.03 ± 0.004 mg/L was obtained from air during the aerosolizing procedure.

In this phase, a significant reduction of total microbial counts was achieved during the 15 min of aerosolization (*p* = 0.026).

After 15 min, microbiological results obtained from air showed total microbial counts at 22 and 37 °C lower than 8 × 10 CFU/m^3^, with a mean value of 1.5 × 10 ± 1.1 × 10 CFU/m^3^. A mean total chlorine value of 0.06 ± 0.002 mg/L was obtained in air after the aerosolizing procedure. In this phase, a significant reduction of total microbial counts was observed in post-aerosolization air samples compared to microbial loads detected during and before the aerosolization procedure (*p* < 0.001).

Waterborne pathogens such as *Legionella* spp. and *Pseudomonas aeruginosa* were not isolated from any air samples.

Figure 2 represents the trend of total microbial counts at 22 and 37 °C, and total chlorine values were detected in all dental units in the three air sampling conditions.

## 4. Discussion

Literature data assert the importance of microbiological risk assessment in dental clinic, which is often due to waterline contamination. Some studies have reported that the primary DUWLs colonizers are not oral bacteria, but rather microorganisms that are normally found in drinking water, such as *Legionella* spp. and *Pseudomonas aeruginosa* [23,24]. Pathogen persistence in DUWLs is usually due to biofilm maturation in plastic tubing, which leads to an unacceptably high number of bacteria in the water used for dental ultrasonication procedures, spraying, cooling, etc. These dental treatments may produce aerosols which can be inhaled by people, including immunocompromised patients with high risk factors for the development of a waterborne infection [25].

Moreover, infectious risk has also been described for healthcare workers involved in dental activities, mostly for *Legionella* spp., which caused a fatal Legionnaires’ disease case [26].

These data highlight the importance of environmental disinfection in dental practices in order to achieve the total compliance of hygiene requirements in all environmental matrices (water, air, and surfaces), mostly in dental clinics dedicated to oral surgical practices (dental clinic 5). Considering the difference between infectious risk in dental and surgical clinics, our premises are organized in order to subdivide all ambulatory practices (orthodontics, dental hygiene, conservative dentistry, etc.), performed in dental clinic 1–4, from surgical activities (oral surgery, salivary gland biopsy, etc.), performed in dental clinic 5.

Indoor microorganisms can originate from the outdoor air and anthropogenic sources, such as building occupants and their activities. Italian guidelines [27] and further literature data [28] consider up to 20 CFU/m^3^ as acceptable values for airborne total microbial load in operating setting, having a unidirectional flow aeraulic system and values up to 100 CFU/m^3^ as an acceptable value in dental offices.

More recent standards recommend the absence of waterborne pathogens (*Legionella* spp. and *Pseudomonas* spp.) and the importance of microbiological risk evaluation in dental premises during the COVID-19 pandemic [29]. However, international standard confirms that dental unit water used in nonsurgical procedures should have a total microbial count purity level equal to or less than ≤500 CFU/mL [30].

Disinfection compliance may be achieved with different chemical biocides suggested by guidelines for drinking water quality [31], which describe the effectiveness of chlorine-based compounds for water disinfection in different plants, including DUWLs.

The synergistic activity of chlorine dioxide with water osmosis is an effective method still used for disinfection of some medical devices using water, such as dialysis plant. High level disinfection of water supply integrated to water desalination may prevent waterline colonization at the initial and final steps of the water production chain [32].

In our study, water treatment obtained by O-CD system allowed the absence of waterborne pathogens in all DUWLs and in the whole water network, as recommended by international standards [30]. Moreover, total microbial counts at 22 and 37 °C almost always resulted in 0 CFU/mL. The lack of bacteria presence is due to the desalination procedure before the water entrance in dental unit plants. This activity prevents the entrance of inorganic and organic compounds, which could favor biofilm formation in water pipelines if it is integrated to water chlorination [33,34].

Chlorination procedure ensured a biocidal residual activity in all DUWLs. In water point of use, we detected free and total chlorine values at the same concentrations suggested by the Council Directive 98/83/EC (about 0.2 mg/L) [20].

Considering that the maximum safe limit for free and total chlorine dioxide in drinking water is 0.8 mg/L (U.S. Environmental Protection Agency) [35], our concentrations do not represent any chemical risk for human health (contact, inhalation, and ingestion).

Considering the aerosolizing procedure applied in dental clinics, chlorine residual activity also appears effective against the airborne microbial population. Not far from the aerosolizing source (1 m), in all dental clinics we observed a significant bacteria reduction in correspondence to chlorine concentration increase detected in air. The best bacteria reduction and chlorine increase was mostly achieved after 15 min of aerosolizing activity.

In pre-aerosolizing air samples, we detected mean total microbial counts at 22 and 37 °C ≥ 1 × 10^2^ CFU/m^3^ with the absence of chlorine. After 15 min of chlorinated water aerosolizing, a mean total microbial count at 22 and 37 °C was >50 CFU/m^3^ with a mean chlorine value of 0.06 mg/L (Figure 3). Therefore, we detected a strong negative correlation between the airborne bacteria reduction and chlorine increase (r −0.96; *p* values < 0.001).

## 5. Conclusions

Our results show how the integrated disinfection method based of water osmosis treated with chlorine dioxide may ensure the total compliance in hygiene requirements in dental premises. This strategy is useful to prevent the DUWLs contamination by waterborne pathogens and, at the same time, the chlorine residual activity resulted in being effective against the airborne bacteria after 15 min of aerosolizing. This activity was shown at the patient chair point (1 m from aerosolization source). Considering the mean chlorine values reached in air, O-CD method may prevent the microbiological risk done by environmental bacteria and respiratory viruses transmitted by droplets (SARS-CoV-2) [36].

In conclusion, this study highlights how a combined disinfection method based on water osmosis and chlorination with chlorine dioxide appears effective for water and air treatment after the aerosolizing procedure with chlorinated water, reducing the infectious risk both for patients and professional staff in dental premises.

## Figures and Tables

**Figure 1 ijerph-19-10562-f001:**
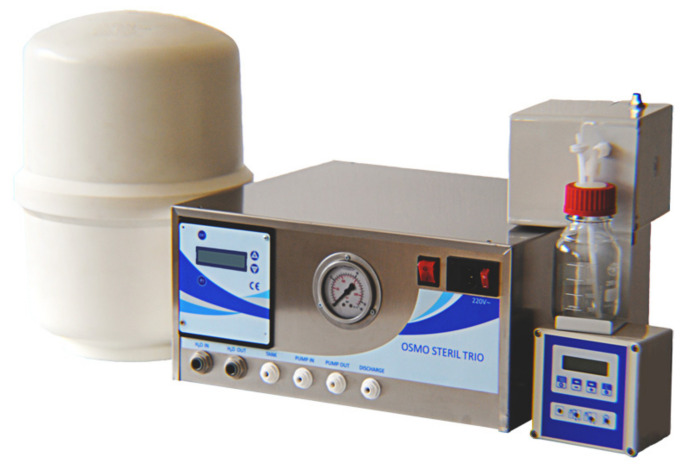
O-CD devices applied in dental practices for drinking water treatment with chlorine dioxide after desalination with reverse osmosis support.

**Figure 2 ijerph-19-10562-f002:**
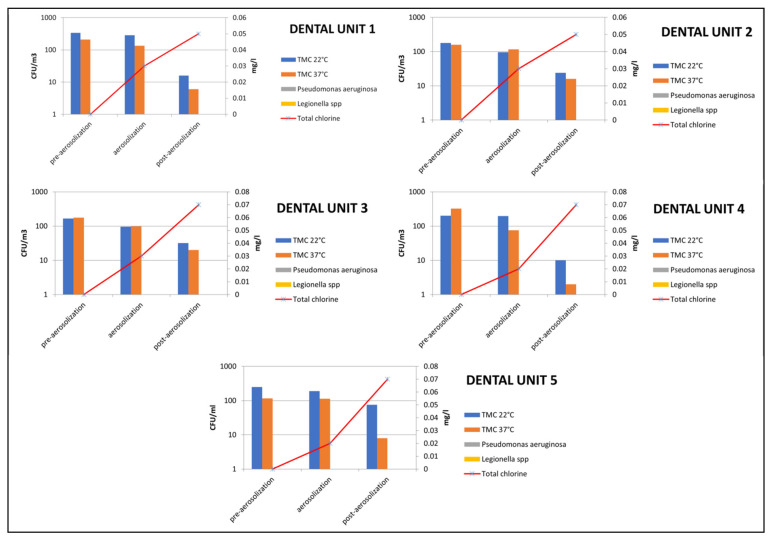
Mean values of Total Microbial Counts (TMC) at 22 and 37 °C, *Pseudomonas aeruginosa*, *Legionella* spp., and mean total chlorine values detected in all air sampling conditions (pre-aerosolization, aerosolization, post-aerosolization procedures).

**Figure 3 ijerph-19-10562-f003:**
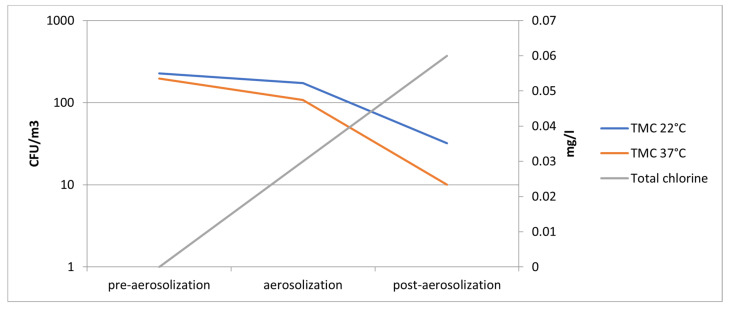
Reduction of mean Total Microbial Counts (TMC) at 22 and 37 °C in correspondence to mean chlorine increase from pre-aerosolizing to post-aerosolizing air samples.

**Table 1 ijerph-19-10562-t001:** Mean values of Total Microbial Counts (TMC) at 22 and 37 °C, *Pseudomonas aeruginosa, Legionella* spp., and mean free and total chlorine values detected in all water sampling points (taps, spittoons, and handpieces).

DENTAL CLINIC 1						
SAMPLING POINT	TMC 22 °C (CFU/mL)	TMC 37 °C (CFU/mL)	*Pseudomonas aeruginosa* (CFU/100mL)	*Legionella* spp. (CFU/l)	Free Chlorine (mg/L)	Total Chlorine (mg/L)
Tap	2 ± 0.9	0	0	<100	0.22 ± 0.07	0.23 ± 0.08
Spittoon	2 ± 1.0	0	0	<100	0.18 ± 0.05	0.22 ± 0.06
Handpieces	0	0	0	<100	0.17 ± 0.09	0.18 ± 0.07
**DENTAL CLINIC 2**						
Tap	0	0	0	<100	0.18 ± 0.06	0.22 ± 0.08
Spittoon	0	0	0	<100	0.18 ± 0.05	0.22 ± 0.04
Handpieces	0	0	0	<100	0.17 ± 0.07	0.18 ± 0.05
**DENTAL CLINIC 3**						
Tap	1 ± 0.02	0	0	<100	0.22 ± 0.05	0.23 ± 0.08
Spittoon	0	0	0	<100	0.20 ± 0.07	0.22 ± 0.06
Handpieces	0	0	0	<100	0.18 ± 0.08	0.23 ± 0.04
**DENTAL CLINIC 4**						
Tap	3 ± 0.05	0	0	<100	0.22 ± 0.07	0.22 ± 0.07
Spittoon	0	0	0	<100	0.22 ± 0.07	0.22 ± 0.06
Handpieces	0	0	0	<100	0.17 ± 0.05	0.20 ± 0.08
**DENTAL CLINIC 5**						
Tap	0	0	0	<100	0.22 ± 0.08	0.22 ± 0.08
Spittoon	0	0	0	<100	0.17 ± 0.07	0.20 ± 0.08

## Data Availability

Data sharing not applicable.

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
