# Peer review of "Effectiveness of a Water Disinfection Method Based on Osmosis and Chlorine Dioxide for the Prevention of Microbial Contamination in Dental Practices"

_ijerph, 2022, doi:10.3390/ijerph191710562_

Round 1
Reviewer 1 Report
The introduction is written in an adequate form. The objectives are well described. In the methods the authors described that the samples were collected in five clinics and that in C5 clinic there are surgical activities.
I suggest that the authors could describe in results as shown in table 1 that in C5 there were no handpieces. In the discussion they could explain about the use of only one clinical with surgical activities and 4 with ambulatorial activities. The main purpose of using these groups in the study
Author Response
Thank you for your comments
I suggest that the authors could describe in results as shown in table 1 that in C5 there were no handpieces.
In results section we added the sentence “Considering the absence of handpieces in dental clinic 5, no data are recorded”
In the discussion they could explain about the use of only one clinical with surgical activities and 4 with ambulatorial activities. The main purpose of using these groups in the study
A reflection related to the ambulatory and surgical activities in our building has been added in discussion section.

Reviewer 2 Report
Thank you for the submission. This was an interesting paper to read. The topic has high clinical relevance. However, I have several comments regarding this submission:
ASTRACT:
1) I would like the pre aerosolizing procedure data to be entered
INTRODUCTION:
2) Please check the layout of the references
3) "Considering this contamination, dental units are recognized as a potential source of infection for human health, especially threatening dentists and patients, who are regularly exposed to water and water-aerosol emitted by aerosol-generating procedures." Add the references and some examples what kind of infections and what risk it is.
4) "The principle of disinfection is characterized by two basic procedures; reverse osmosis applied to the inlet water of the unit (which will be demineralized and free of microbial load) and continuous disinfection of water with chlorine dioxide (which is the real active principle of the system)."Add the reference.
5) Add recent data on CFU concentration with standard methods
M&M
6)What kind of studio is it? Retrospective? Observational? Prospective? IRB registration number? The informed consent of workers within the clinics?
DISCUSSION:
7) Are infectious complications also described for health workers within the clinics?
8) "consider up to 20 CFU/m3 as acceptable values for airborne total microbial load in operating setting having a uniderational flow aeraulic system and values up to 100 CFU/m3 as an acceptable value in dental offices." Add recent data on the concentration of CFU for pseudomonas and and legionella with the standard methods and compare it with the obtained data. Furthermore, what are the risks associated with the concentration of Free Chlorine and total chlorine present in the literature?
I enjoyed reading this manuscript and would be happy to accept with the above comments included.
Thank you and I greatly look forward to the authors' response.
Author Response
Thank you for your comments.
ASTRACT
1) I would like the pre aerosolizing procedure data to be entered
Mean value of total microbial load detected from air before aerosolizing procedure (pre-aerosolizing has been added in abstract
INTRODUCTION:
2) Please check the layout of the references
Check has been done
3) "Considering this contamination, dental units are recognized as a potential source of infection for human health, especially threatening dentists and patients, who are regularly exposed to water and water-aerosol emitted by aerosol-generating procedures." Add the references and some examples what kind of infections and what risk it is.
References and some examples of the most cited infections have been added in introduction section.
4) "The principle of disinfection is characterized by two basic procedures; reverse osmosis applied to the inlet water of the unit (which will be demineralized and free of microbial load) and continuous disinfection of water with chlorine dioxide (which is the real active principle of the system)."Add the reference.
Reference has been added
5) Add recent data on CFU concentration with standard methods
Some data and relative reference about the abatement of waterborne pathogens (E.coli) with chlorine dioxide and relative standard method have been added in introduction section.
M&M:
6)What kind of studio is it? Retrospective? Observational? Prospective? IRB registration number? The informed consent of workers within the clinics?
All information about the type of study and the informed consent have been added in methods section. Considering the absence of patients during the study IRB registration was not was not considered necessary
DISCUSSION:
7) Are infectious complications also described for health workers within the clinics?
This description with relative reference has been added in results section.
8) "consider up to 20 CFU/m3 as acceptable values for airborne total microbial load in operating setting having a uniderational flow aeraulic system and values up to 100 CFU/m3 as an acceptable value in dental offices." Add recent data on the concentration of CFU for pseudomonas and legionella with the standard methods and compare it with the obtained data.
Statements have been enriched with more recent data and reference standards about the CFU limits detectable in DUWLs. The compliance of our microbial data obtained in water to the reference standards has been highlighted in discussion section.
Furthermore, what are the risks associated with the concentration of Free Chlorine and total chlorine present in the literature?
All information and relative reference have been added.
